# Integrating Omics Technologies for a Comprehensive Understanding of the Microbiome and Its Impact on Cattle Production

**DOI:** 10.3390/biology12091200

**Published:** 2023-09-01

**Authors:** Harpreet Kaur, Gurjeet Kaur, Taruna Gupta, Deepti Mittal, Syed Azmal Ali

**Affiliations:** 1Division of Biochemistry, ICAR-National Dairy Research Institute (ICAR-NDRI), Karnal 132001, India; 2Centre for Healthy Brain Ageing, School of Psychiatry, University of New South Wales, Sydney, NSW 2052, Australia; 3Mark Wainwright Analytical Centre, Bioanalytical Mass Spectrometry Facility, University of New South Wales, Sydney, NSW 2052, Australia; 4Steno Diabetes Center Copenhagen, DK-2730 Herlev, Denmark; 5Cell Biology and Proteomics Lab, Animal Biotechnology Center, ICAR-National Dairy Research Institute (ICAR-NDRI), Karnal 132001, India; 6Division Proteomics of Stem Cells and Cancer, German Cancer Research Center, 69120 Heidelberg, Germany

**Keywords:** cattle, dairy, metabolomics, microbiota, ruminant, system biology

## Abstract

**Simple Summary:**

This review article highlights the pivotal role of ruminant production in global agriculture and the challenges posed by population growth. The focus on refining ruminant production systems is in-creasing due to environmental concerns. Recent investigations emphasize the link between the rumen microbiome composition and economically advantageous cattle traits, driving the need for innovative strategies to enhance feed efficiency while reducing environmental impact. Omics technologies offer fresh insights into metabolic health changes in dairy cattle, enhancing nutri-tional management. The key role of the rumen microbiome in improving feeding efficiency by converting low-quality feed into energy substrates is underscored. This microbial community is vital in gut microbiome studies, contributing to plant fiber digestion and influencing ruminant production and health. Factors like compromised animal welfare can impact rumen microbiology and production rates. A comprehensive global approach targeting cattle and their rumen microbi-ota is essential for feed efficiency and fermentation processes. The review focuses on the factors influencing rumen microbiome establishment after perturbations and host-microbiome interac-tions, highlighting practical applications in animal health and production. Scrutinizing the micro-biome’s effects on cattle production fosters more sustainable food systems, reducing environmental impact. In essence, the review emphasizes the significance of the rumen microbiome in enhancing ruminant performance and explores the complex interplay between microbes, hosts, and their en-vironment to achieve sustainable and efficient livestock production.

**Abstract:**

Ruminant production holds a pivotal position within the global animal production and agricultural sectors. As population growth escalates, posing environmental challenges, a heightened emphasis is directed toward refining ruminant production systems. Recent investigations underscore the connection between the composition and functionality of the rumen microbiome and economically advantageous traits in cattle. Consequently, the development of innovative strategies to enhance cattle feed efficiency, while curbing environmental and financial burdens, becomes imperative. The advent of omics technologies has yielded fresh insights into metabolic health fluctuations in dairy cattle, consequently enhancing nutritional management practices. The pivotal role of the rumen microbiome in augmenting feeding efficiency by transforming low-quality feedstuffs into energy substrates for the host is underscored. This microbial community assumes focal importance within gut microbiome studies, contributing indispensably to plant fiber digestion, as well as influencing production and health variability in ruminants. Instances of compromised animal welfare can substantially modulate the microbiological composition of the rumen, thereby influencing production rates. A comprehensive global approach that targets both cattle and their rumen microbiota is paramount for enhancing feed efficiency and optimizing rumen fermentation processes. This review article underscores the factors that contribute to the establishment or restoration of the rumen microbiome post perturbations and the intricacies of host-microbiome interactions. We accentuate the elements responsible for responsible host-microbiome interactions and practical applications in the domains of animal health and production. Moreover, meticulous scrutiny of the microbiome and its consequential effects on cattle production systems greatly contributes to forging more sustainable and resilient food production systems, thereby mitigating the adverse environmental impact.

## 1. Introduction

Microorganisms play a pivotal role in fundamental biogeochemical processes, including nitrogen and carbon cycles, as well as in crucial biological functions such as waste decomposition [1,2,3,4,5]. Moreover, these microorganisms constitute a substantial proportion of cellular entities within organisms and fulfill indispensable roles for both global ecosystems and individual well-being [1,2,3,4,5]. Among the earliest livestock species, ruminants possess a distinct ability to transform low-quality forage into high-value meat and dairy products. Over time, extensive research efforts have been dedicated to enhancing the productivity of ruminant livestock. Recent strides in advanced technologies, such as next-generation sequencing and mass spectrometry, have catalyzed remarkable progress in the investigation of the rumen microbiome. These innovations have facilitated the complete genome sequencing and comprehensive profiling of the rumen microbial community [1,2,3,4,5,6,7,8,9].

The rumen creates an environment that supports a dense and diverse community of anaerobic microorganisms. These microorganisms serve a crucial metabolic function by breaking down complex plant polysaccharides, a task that ruminants are unable to perform on their own. The microorganisms within the rumen play diverse roles in the digestion of feed, engaging in the fermentation of both structural and nonstructural carbohydrates and proteins from plants. This process yields volatile fatty acids, which serve as primary nutritional sources for the host animal and play a substantial role in enhancing ruminant productivity. Within the gastrointestinal tract (GIT) of ruminants like cattle, the rumen and the lower GIT accommodate a range of symbiotic, commensal, and parasitic microorganisms. These microorganisms exert significant effects on the physiology and performance of the animals, thereby contributing to the overall metabolic capacity of the rumen [8,9,10].

Researchers have extensively explored the microbiomes within the rumen and lower gastrointestinal tract of cattle, specifically focusing on production-related traits like feed efficiency. This investigation aims to enhance the comprehension of the intricate relationship between host organisms and their microbial inhabitants. The microbiome exerts a pivotal influence on various aspects of host metabolism, physiology, and immunity, playing an indispensable role in diverse stages of animal health [10]. Gastrointestinal microbes hold a pivotal position in the health and disease dynamics of animals. They significantly contribute to the development of the immune system, the synthesis of bioactive compounds, and the efficient extraction of energy, among other essential functions. However, the field of microbiology faced limitations for an extended period due to inadequate research tools and the challenge of cultivating and studying a substantial portion of the microbial population residing within the gastrointestinal tract. The mutualistic bond between microbiota and their host gives rise to an array of advantageous microbial products, such as short-chain fatty acids (SCFAs), secondary bile acids, and vitamins, all of which confer benefits upon the host [11,12]. Animal husbandry plays a vital role in the provisioning of human sustenance and constitutes a significant cornerstone of the agricultural economy. Moreover, animals possess the ability to digest biomass that is inaccessible to humans, thereby offering a solution to the competition for food resources between human populations and livestock. For instance, ruminants possess the capacity to utilize plant cell wall biomass, agricultural by-products, and waste generated by the food industry as viable sources of nutrition, thanks to the role of the rumen microbiota [13].

Rumen microorganisms can be categorized into distinct functional clusters, encompassing cellulolytic, amylolytic, proteolytic, and other groups. These clusters engage in the degradation of a broad spectrum of feed components and also process certain byproducts generated by other microbial entities [1,14]. Within this microbial consortium, methanogens, a type of archaea responsible for methane production, play a role in metabolizing hydrogen produced by specific fermentative microorganisms, leading to the formation of methane. This fermentation process is linked to the emission of methane, a significant contributor to anthropogenic greenhouse gas emissions, and results in a loss of energy from the animal’s feed [14]. Consequently, a comprehensive understanding of rumen microbial communities is imperative to fathom the intricate conversion of plant materials within the rumen into both unwanted and valuable ruminant-derived products [15]. The ramifications of methane production have been scrutinized across diverse sectors, ranging from agriculture and the environment to the realm of biomedicine [16]. Notably, the environmental consequences stemming from the beef and dairy cattle industries, which play a substantial role in methane emissions, are under intense scrutiny, even as the global demand for meat and dairy products continues to surge. Given that methane production leads to energy dissipation, novel and inventive strategies are being devised to curtail methane emissions from livestock, boost the efficiency of ruminant livestock production, and diminish the associated carbon footprint (Figure 1).

Looking forward, an inclusive research approach that encompasses diverse animal species, varied dietary strategies, and a spectrum of animal ailments could yield substantial implications for both animal agriculture and health. There exists a compelling need to delve deeper into the interplay between microorganisms, host mucosal surfaces, and the host itself. This pursuit is particularly crucial as the bacterial communities associated with hosts undergo continuous transformations across generations and are correlated with diseases linked to aging [17,18]. Notably, the epithelial mucus layer on the mucosal surface is considered a primary niche for host-associated bacteria. However, it is plausible that the potential contributions of other pivotal members within the microbial consortium, such as archaea, fungi, and viruses, might still be underestimated. Therefore, the amalgamation of traditional culture methodologies with advanced molecular tools will prove pivotal in extending our comprehension of these often overlooked microorganisms, thereby fostering a comprehensive and functional grasp of the overall meta-organism [19,20]. The emergence of high-throughput DNA sequencing techniques, complemented by novel advancements in bioinformatics, has facilitated the identification and characterization of microbes and their genetic components (microbiome) residing within and on the body. This confluence of methodologies has substantially augmented our existing knowledge within this realm.

This review serves to accentuate contemporary investigations concerning the microbiomes within the rumen and hindgut, specifically within the context of livestock performance. Additionally, it centers its attention on elucidating the intricate connections that exist between the host-ruminal microbiome and its repercussions on both animal productivity and host well-being. The advent of the meta-omics revolution, characterized by advanced sequencing techniques and their corresponding analytical frameworks, has ignited a surge of enthusiasm for comprehending the ways in which the microbiome influences physiological processes and the vulnerability to various diseases.

## 2. Multi-Omics Approaches for Understanding Microbe-Host Interactions: Implications for Animal Health and Nutrition

The gastrointestinal tract, whether in animals or humans, undergoes significant influence due to the coexistence and interplay of a myriad of host cells, microbial entities, metabolites, and secreted proteins, collectively exerting an impact on overall health. The amalgamation of advanced high-throughput multi-omics technologies with the study of microbiota within ruminant livestock presents an opportunity to delve profoundly into the intricate dynamics of the microbe-host relationship, as well as to assess the repercussions of dietary strategies on animal performance [21]. Across recent decades, thorough investigations have brought to light the pervasiveness and intricate nature of microbiota, thereby substantiating the pivotal role played by microbial communities.

Methods that do not rely on culturing, collectively known as culture-independent approaches, have gained significant traction. An initial example of such an approach was the pioneering use of low-throughput bacterial sequencing, which centered around the 16S ribosomal RNA gene sequencing method [22]. This method’s advent marked a turning point in sequencing techniques, rendering them more accessible and potent. In tandem with this, advancements in genome-wide sequencing methodologies have emerged, including metagenomics, metatranscriptomics, metaproteomics, and metabolomics. These advancements have furnished the scientific community with supplementary tools to unravel the intricacies of microbiome functions at the molecular level. These multi-omics strategies delve into the genetic blueprints, RNA transcripts, proteins, and metabolites emanating from a multitude of microorganisms, facilitating an analysis of system-level processes and the interactions among these microbial entities [21,22].

Metagenomics, often known as shotgun sequencing, DNA-seq, or RNA-seq, stands as a culture-independent sequencing methodology that delves into the entire nucleotide repertoire present within a community, extracted from clinical or environmental samples (Figure 2). Extensive investigations reliant on metagenomics have furnished valuable insights into the expanse of diversity inherent to the microbiome. Furthermore, these studies have pinpointed prospective targets with clinical implications for associated diseases [23,24]. Despite the elevated costs and longer run durations associated with metagenomics in comparison to the more streamlined 16S ribosomal RNA sequencing method, the comprehensive and intricate information garnered through this approach eclipses the additional investments in resources.

Nonetheless, it is important to note that the presence of a gene within the metagenome does not necessarily indicate its expression, nor does it offer insights into the specifics of its expression patterns. To bridge this gap, metatranscriptomics sequencing comes into play, enabling the monitoring of alterations in microbial gene expression over time. This becomes particularly relevant when investigating shifts within the microbiota as they respond to disturbances or changes in their environment [25]. However, metatranscriptomics poses a significant challenge, mainly revolving around the intricacies of sample preparation. Gene expression profiles can undergo swift changes due to the relatively low stability of mRNA during the preparation process. Consequently, the pool of recovered mRNA might portray the microbiome’s expression pattern under the stress conditions induced by the act of sampling itself.

Metaproteomics emerges as a promising avenue for conducting comprehensive investigations encompassing the entirety of the proteome present within clinical or environmental samples. This technique serves to delve deeply into the functional diversity inherent to the microbiota. Through enabling the dissection of protein-protein interactions, molecular networks, and post-translational modifications, as well as revealing the microbiota’s responses to external stressors and disease conditions, metaproteomics presents itself as a more robust method that provides a truer reflection of the microbial ecosystem in comparison to metatranscriptomics [26].

Furthermore, the intrinsic stability of proteins compared to RNA offers a distinct advantage during the process of sample preparation. Metaproteomics holds the capacity to furnish more intricate insights into the metabolic processes orchestrated by the microbiota. This is a facet where the study of RNA expression often falls short, as it does not encompass the realm of post-transcriptional regulation. In the metaproteomic workflow, proteins are segregated and subjected to digestion via proteolytic enzymes. Subsequently, liquid chromatography is directly coupled with mass spectrometry analysis, following meticulous optimization of the sample preparation protocol to mitigate the inadvertent co-extraction of eukaryotic proteins [27].

Contemporary metagenomic explorations into the microflora across diverse mammalian species have unveiled noteworthy findings. Specifically, these studies have demonstrated that anabolic pathways responsible for amino acid (AA) synthesis are more prominently prevalent within the microflora of ruminants and herbivores compared to carnivores. This divergence arises from the dietary variations: carnivores are endowed with a protein-rich diet, resulting in a microbiota that exhibits heightened proteolytic activity. In contrast, herbivores rely on a diet rich in fiber, where carbohydrates serve as the primary energy source [28].

The identification of proteins is executed by referring to an accessible sequence database, either derived from public resources or tailored to the specific samples under study [29] (Table 1).

In recent times, the field of microbial ecology has increasingly gravitated towards the adoption of metabolomics technologies. This shift is largely attributed to remarkable technological progress that now enables the extensive analysis of thousands of metabolites (Figure 2). Leveraging high-throughput methodologies, such as Nuclear Magnetic Resonance (NMR) and Mass Spectrometry (MS)-based techniques, in conjunction with advanced bioinformatics tools, holds the potential to unravel the intricate metabolic imprints of host animals. A comprehensive review by Peters et al. in 2019 delves deeply into this area, providing a detailed exploration of these methodologies [44].

Nevertheless, despite the promise held by meta-omics integration, the execution of such studies remains constrained by factors like sample size limitations, annotation rates, and the depth of sequencing. Consequently, only a handful of investigations have successfully achieved such integration. Moreover, the interpretation of data remains a critical concern. The datasets available are not immune to imperfections stemming from technical biases and sampling insufficiencies. Historically, research into the microbial composition of ruminant livestock has lagged behind that of human microbiota. Nonetheless, the unveiling of immediate impacts achievable through meta-omics approaches should garner substantial interest for future inquiries, particularly in the context of refining animal nutrition strategies and enhancing animal health.

## 3. The Complex and Vital Role of Rumen Microbiota in Ruminant Nutrition and Health

The bovine rumen encompasses a complex and distinct microbial assembly, characterized by its phylogenetic diversity. This microbial community comprises anaerobic bacteria, fungi, methanogenic archaea, ciliate protozoa, and viruses. Among these constituents, bacteria dominate the rumen microbial consortium, with a concentration ranging from 10^10^ to 10^11^ cells/mL of rumen fluid. This bacterial population encompasses phyla such as Firmicutes, Bacteroides, and Proteobacteria, which play pivotal roles in the breakdown of a wide spectrum of dietary polysaccharides and peptides [45,46].

Rumen archaea, in contrast, are found in numbers ranging from 10^6^ to 10^8^ cells/mL of rumen fluid. These archaea are primarily affiliated with the Euryarchaeota phylum, with the Methanobrevibacter ruminantium and Methanobrevibacter gottschalkii clades emerging as the most prevalent [45,46]. The rumen’s protozoan inhabitants are present at a density spanning 10^4^ to 10^6^ cells/mL of rumen fluid, with dominant genera including Entodinium, Polyplastron, Epidinium, and Eudiplodinium [45]. Rumen fungi contribute to the microbial composition with a population range of 10^3^ to 10^6^ zoospores/mL of rumen fluid. These fungi exclusively belong to the Neocallimastigomycota phylum [45]. The rumen virome, mainly composed of Caudovirales [47], is influenced by a multitude of factors, prominently including ruminant species, diet, age, gender, environment, and geographical location [9,48]. This microbial community showcases a significant degree of functional redundancy. It encompasses cellulolytic, hemicellulolytic, amylolytic, proteolytic, and lipolytic species, collectively endowed with the capacity to transform otherwise indigestible plant biomass into valuable energy resources for the host (Figure 3).

Across the animal kingdom, ranging from invertebrates to vertebrates, a diverse and densely populated resident microbial community is a constant presence. This community encompasses bacteria, archaea, protists, viruses, and fungi, predominantly residing in mucosal organs like the oral cavity and the intestine. This internal microbial assembly within the host is referred to as the microbiota [49].

Ruminants are inherently reliant on their microbiota to facilitate feed digestion, a relationship that directly influences their survival. This interdependence has prompted researchers to explore potential correlations between the abundance and composition of rumen microbial taxa and the manipulation of host physiological parameters [30]. In essence, the rumen functions as an anaerobic fermentation chamber prior to the stomach, specifically designed for the breakdown of indigestible dietary constituents. Notably, certain ingested carbohydrates—such as cellulose, hemicellulose, inulin, and starch—undergo hindered digestion due to the absence of requisite digestive enzymes [50]. Consequently, these recalcitrant carbohydrates traverse towards the latter parts of the gastrointestinal tract (GIT), where they serve as substrates for microbial fermentation.

Within the microbiota, key members undertake the hydrolysis of ingested fiber, sugars, protein, and lipids, transforming them into shorter chains (oligomers) and individual molecules (e.g., glucose, and amino acids). These products then serve as substrates for distinct members of the microbial ensemble [47]. Notably, the central outputs of microbial fermentation within the rumen encompass volatile fatty acids (VFAs), with acetate, propionate, and butyrate prevailing. These VFAs are produced by utilizing dietary fibers, carbohydrates, and proteins as primary substrates. Ruminants derive around 70% of their daily energy requisites through the fermentation orchestrated by the rumen microbiota, acting on otherwise indigestible feed components. As a result, studies investigating the microbiome of bovines have predominantly focused on the rumen microbiota, given its substantial impact on the host’s physiological parameters, particularly in relation to milk production [48].

The rumen microbiome assumes a pivotal role in shaping host physiology, and any disruptions within this ecosystem have the potential to exert a profound impact on host metabolism and overall bodily composition [1]. Consequently, delving into the contribution of these microorganisms to ruminant production has garnered significant attention. This is driven by an acknowledgment of the pivotal role that rumen microbe-driven metabolic processes play in ensuring the host’s well-being [30,49]. The composition of the rumen microbial community is intricately linked to a range of factors, including feed efficiency, methane emissions, health status, and milk composition. This microbial composition further influences the maintenance of health homeostasis, which holds immense significance for the optimal performance of animals in the production realm [50,51,52,53].

In the realm of animal production industries, the well-being of the is intricately intertwined with overall animal health, bestowing a multitude of physiological and functional advantages upon the host [1]. Within this context, the rumen microbiota emerges as a vital contributor, assisting in various aspects such as food processing, the breakdown of nutrients, vitamin synthesis, and bolstering the development of the host’s immune system. These benefits are achieved through mechanisms like competition with pathogenic microbes to thwart their harmful colonization and propagation. Furthermore, the rumen microbiota contributes to energy generation, notably through the production of volatile fatty acids (VFAs) from complex carbohydrates that remain indigestible by the host [54]. Among these roles, the production of VFAs holds particular prominence. VFAs make a significant contribution by fulfilling a substantial portion of ruminants’ energy requirements. Additionally, they play a pivotal role in the development of the rumen, especially in the formation of rumen papillae. As a result, VFAs become indispensable to animal performance, ensuring optimal growth and productivity [55].

The rumen microbiota exerts a notable influence on pre-ruminant management, with a particularly profound impact on the weaning process. This influence stems from its indirect involvement in rumen development, which, in turn, relies on the fermentation of complex carbohydrates carried out by these microorganisms [40]. Additionally, the rumen microbiota plays a key role in metabolizing nitrogen-containing compounds. This includes processing peptides, ammonia, and urea, all of which assume a critical role in allocating microbial proteins to the host. This allocation, in turn, contributes to the development of milk production and muscle growth within the host [56].

The microbiota extends its positive influence to the intestinal epithelium, where it plays a significant role in promoting the growth of intestinal microvilli. Additionally, it has been demonstrated to hold a crucial role in modulating both innate and adaptive immune responses. This interaction is orchestrated through a diverse array of signaling molecules and metabolites derived from the microbiota. This complex interplay ultimately leads to the establishment of tolerance towards a multitude of microbial antigens [57]. The maintenance of intestinal integrity stands as a paramount concern in the growth and development of food animals. A robust intestinal barrier aids in curbing the chronic inflammation that might arise from dietary sources. Such inflammation could potentially prove fatal to animals. Furthermore, an intact intestinal barrier supports the establishment of an immunologically normal and steady state within the gut throughout the animals’ lifespan [58].

The rumen microbiota showcases a remarkable capability to produce and regulate a spectrum of hormonal products and bioactive peptides, thereby exerting an influence on gut health dynamics [32]. These products, upon entering the bloodstream for subsequent distribution, wield an impact on the functioning of neighboring organs and systems [59]. This process introduces a range of dynamics that hold implications for the growth and development of ruminants. Notably, the intricate hormonal balance potentially holds significant promise for facets like neuronal development. For instance, this hormonal homeostasis could potentially contribute to mental health outcomes during the early stages of life and impact stress responsivity in adulthood [60]. In light of its extensive array of metabolic activities and its regulatory capacity, coupled with its potential to influence the development and functioning of distant organs, considering the microbiota as a “virtual organ” offers a conceptual framework to unravel its intricate and multifaceted role [61,62].

In summation, it is evident that the intricate arrangement of the rumen microbial community serves a pivotal function in upholding the immunity and overall physiology of ruminants [63]. Yet, despite the myriad significant roles fulfilled by the rumen microbiota in the context of bovine health and production, a comprehensive understanding of the underlying mechanisms remains somewhat elusive [64].

## 4. Factors Affecting Microbiome Establishment in Rumens

Within the ruminant intestines, a diverse array of microbiota predominates, predominantly comprised of bacteria with a specialized role in breaking down intricate nutrients like cellulose and hemicellulose. This breakdown results in the conversion of these complex compounds into simpler constituents, such as glucose, which ultimately facilitates the assimilation of nutrients [47].

Maintaining proper homeostasis of the rumen microflora stands as a crucial aspect. Their presence in optimal quantities is indispensable, as they exert a substantial influence on the host’s physiological state and play a notable role in methane production [63,65]. The composition of these microbial communities within the rumen is contingent on a multitude of factors. These include breed, age, external environment, diet, and nutritional factors (as outlined in Table 1) [9,42]. Moreover, other variables, such as the process of weaning, energy requirements, and the presence of potentially toxic metabolic byproducts, can also exert an impact on the microbial composition and abundance within the rumen. External factors, including heat stress, psychological stress, environmental conditions, and dietary patterns, can likewise pose threats to the stability of rumen microflora [66].

Amidst the multitude of influential factors, diet emerges as a prominent determinant that governs the composition of the microbiota. This role is particularly evident in the context of the temporal changes occurring in the rumen microbiome of neonatal calves [30,67]. For instance, consider the rumen component of three-week-old calves fed with milk replacer, alongside supplementation of a neonatal crude starter ration containing 20% protein, 3% fat, and 5.7% fiber. In this scenario, a relatively uniform representation of Prevotella (15.1%) and Bacteroides (15.8%) is observed. In contrast, calves exclusively fed a milk replacer diet exhibit an altered rumen microbiota. This shift involves a transition from the dominant presence of Prevotella to Bacteroides within the initial six weeks of their lives [31]. Consequently, the concurrent prevalence of Prevotella and Bacteroides in 3-week-old calves that are supplied with milk replacer alongside a calf starter diet suggests a plausible connection between the age-dependent shift in dominant bacteria, such as Prevotella, and the introduction of fiber-rich dietary supplements. This underscores the pivotal role that diet plays in influencing the trajectory of rumen microbiota development in neonatal calves.

The establishment of a well-functioning rumen microbiota is of utmost importance, and an effective diet plays a vital role in facilitating the process. A well-designed diet aids in directing milk to bypass the rumen and enter the abomasum, a critical process for the digestive physiology of ruminants [40]. This mechanism ensures proper nutrient absorption and utilization.

Furthermore, it is noteworthy that the composition of the pre-weaning diet and the methods of feeding exert a substantial influence on the rumen microbial community. These factors can even shape the composition and density of methanogens, bacteria, and protozoa in the rumen after the weaning process in pre-weaned lambs [68,69,70]. This highlights the significance of closely monitoring pre-weaning feeding practices, as these factors have a lasting impact on the development of microbial fermentation capacity and the establishment of the rumen microbiota [60]. Consequently, it is clear that managing both pre-weaning and post-weaning feeding strategies is integral for fostering a robust microbial fermentation capacity and cultivating a well-balanced rumen microbiota (as depicted in Figure 1 and Figure 3).

The type of feed that ruminants consume plays a pivotal role in shaping the composition of their microbiota, and this, in turn, exerts a direct impact on the levels of methane production [71,72,73]. Notably, the presence and abundance of specific bacteria, such as Prevotellaceae, which are proficient in producing propionate, substantially influence the level of methane production. This aspect underscores the importance of maintaining a well-balanced feed ration, particularly in terms of the ratio between roughage and concentrate components. The balance in this ratio holds a significant bearing on methane emissions [72,73]. Specifically, a diet with a higher proportion of roughage tends to result in elevated methane production, while a diet rich in concentrated feed leads to comparatively lower methane emissions.

It is noteworthy that among the constituents of animal feed, crude fiber stands out as the most methanogenic component, driving methane production. Conversely, the presence of crude fat in the diet tends to have an opposing effect. This emphasizes the necessity of ensuring a properly balanced diet to avert potential fertility-related disorders and to maintain optimal milk production levels in high-yielding cows [74]. In essence, the intricate relationship between diet, microbiota composition, and methane production underscores the significance of a carefully tailored feeding strategy in ruminant husbandry.

Furthermore, the balance between roughage and concentrated feed within the housing system is critical. An imbalance in this aspect, coupled with frequent and abrupt changes in food portions and short adaptation times, has the potential to disrupt the functioning of the digestive system’s microbiota. This disruption, in turn, has a cascading effect on the overall health and productivity of the animals [66,74]. A pertinent study involving the analysis of rumen fluid collected from animals across diverse feeding practices revealed compelling insights. It was observed that a pasture-based diet led to an increase in the population of bacteria belonging to the Bacteroidetes class, specifically the Ruminococcaceae family. In contrast, a diet predominantly based on cereals caused a rise in the abundance of bacteria from the Prevotellaceae and Succinivibrioaceae (Proteobacteria) groups. Remarkably, these trends held consistent regardless of the specific ruminant species under scrutiny [75]. This underscores the profound impact that diet choices exert on the microbial communities residing within the rumen, with potential repercussions on animal health and productivity.

Transitioning from a low-energy diet to a high-energy feed can have substantial repercussions on the functionality of the digestive system and the microbiome. Such transitions can disturb the delicate equilibrium, leading to a disruption in the breakdown of nutrients from the feed. To mitigate this, it is crucial to adjust the feed volume accordingly. This is due to the fact that, in such instances, the microbiome might struggle to comprehensively break down the biomass present in the feed. Consequently, this impaired microbial breakdown results in a reduced capacity for nutrient absorption by the host’s digestive system [74,76]. This emphasizes the need for careful dietary management and gradual transitions to avoid upsetting the microbiome’s delicate balance and ensuring optimal nutrient utilization by the host.

### 4.1. Age-Dependent Changes in Microbial Population

The establishment of the rumen microbiota during the neonatal stage of ruminants holds paramount importance for their proper development. Notably, anaerobic microorganisms rapidly become prevalent within the rumen, with their predominance setting in as early as the second day of life. By this point, their density reaches an impressive count of 10^9^ colony-forming units (CFU) per milliliter of rumen fluid. Additionally, the population of cellulolytic bacteria stabilizes at around 10^7^ CFU/mL of rumen fluid within the first week of life [77].

Importantly, it is pertinent to acknowledge that the dominant bacterial species found in the rumen of neonatal lambs differs from those prevalent in adult ruminants. The microbial landscape of the neonatal rumen also includes other key groups, such as anaerobic fungi and methanogens, which begin colonizing the rumen between 8 and 10 days after birth. Protozoa, on the other hand, make their appearance only after 15 days postpartum [78]. This intricate timeline underscores the gradual and dynamic development of the rumen microbiota in neonatal ruminants, contributing significantly to their physiological maturation.

A comparative analysis between conventionally reared and conventionalized lambs has illuminated a crucial aspect in the establishment of the rumen microbiota. Specifically, the presence of a robust and well-established bacterial community serves as a prerequisite for the subsequent colonization of protozoa. This intriguing sequence reveals that protozoa tend to emerge only after the establishment of bacteria within the rumen [77,79]. In the initial stages of rumen microbiota establishment, certain genera take center stage. Notably, the Propionibacterium, Clostridium, Peptostreptococcus, and Bifidobacterium genera assert their dominance. Furthermore, among the cellulolytic bacterial community, Ruminococcus species emerge as the prevalent players [63]. It is been observed that limited exposure to the dam or other animal species during the first few weeks of life can significantly delay the establishment of cellulolytic bacteria. This highlights the pivotal role that early environmental interactions and exposures play in shaping the development of a microbiota that is tailored to the specific host [77]. This underscores the intricate interplay between environmental factors and microbial colonization in the establishment of the rumen microbiota.

### 4.2. Stress-Related Changes in the Composition of the Microbiota

Heat stress is a multifaceted phenomenon influenced by a combination of factors, encompassing ambient temperature, relative humidity, solar radiation, and air movement. It gives rise to a range of noticeable symptoms, including elevated body temperature, accelerated breathing rate, decreased feed consumption, and heightened water intake. The ramifications of heat stress extend to reduced animal performance, and it also exerts an impact on the microflora present in their systems. When faced with heat stress, ruminants exhibit a response by curtailing their intake of dry matter. This serves the purpose of diminishing metabolic heat production, thereby facilitating the maintenance of a stable body temperature [78].

Among ruminants, dairy cattle stand out as particularly vulnerable to the effects of heat stress. This vulnerability is accentuated by the additional demands posed by milk production. As the specter of global warming looms larger, the prevalence of heat stress is increasingly becoming a pressing concern, not only for dairy cattle but for animals across the spectrum, including ruminants (as depicted in Figure 4). This underscores the far-reaching consequences of heat stress on animal well-being and productivity in the context of changing climatic conditions.

Feeding ruminants a diet heavily skewed towards concentrate feed rather than roughages can precipitate a condition known as acidosis, which has disruptive effects on the rumen fermentation process. This shift in diet composition has discernible repercussions on the microbial composition within the rumen. For instance, the population of Fibrobacter and Oscillospira bacteria tends to decline under such circumstances, while there’s an increase in the prevalence of Clostridium coccoides and the Streptococcus/Lactococcus genera. Moreover, there is a discernible alteration in the production levels of short-chain fatty acids (SCFA). Acetate levels and SCFA production diminish, while propionate and lactate levels see an elevation. This shift can be attributed to the presence of Streptococcus bovis, a bacterium known for lactate production. Lactate is less readily absorbed by the rumen epithelium compared to other compounds like acetate, propionate, or butyrate. This, in turn, contributes to a drop in rumen pH, which can range between 6.8 to 6.5 [80].

Notably, humidity levels also factor into this process. Higher humidity levels tend to exacerbate the decline in pH. Furthermore, under conditions of heat stress, escalated water consumption can impede the flow of food content, resulting in prolonged retention within the rumen and a subsequent elevation in rumen fluid acidity. This shift in conditions also has implications for microbial populations. It has been observed that heat stress can prompt an increase in the number of Bacteroidetes and the Spirochetes phylum, accompanied by a decrease in the Firmicutes population [81]. These insights underline the intricate interplay between dietary composition, environmental factors, microbial dynamics, and the resulting impact on rumen health and functioning.

## 5. Role of Lower-Gut Microbiome in Host Gut Health

The gastrointestinal (GI) tract, encompassing both the rumen and the lower gut of cattle, harbors a rich and diverse microbiome responsible for the digestion and fermentation of feed, a process that profoundly influences feed efficiency. The lower gut refers to the post-gastric portion of the intestinal tract, encompassing both the small intestine and the hindgut segments. Within this intricate system, distinct microbiota reside in the rumen and lower gut, each wielding a crucial role in shaping host health (as summarized in Table 2) [47,82,83].

Comprehending the microbiome and its intricate connection with host health has been a focus of research for many years. Traditional cultivation-based methods provided insights into key metabolic pathways. However, molecular biology techniques have significantly broadened our understanding of the complexity of the rumen and intestinal microbiomes in cattle. Among these techniques, metagenomics has risen to prominence as the leading technology for unraveling the intricacies of the GI microbiome and its interplay with host nutrition and health [82].

Microorganisms dwelling within the lower GI tract of cattle and other ruminants wield considerable influence over animal physiology and performance. Earlier investigations have concentrated on unveiling the symbiotic relationships between hosts and microbes within cattle by exploring the microbiomes of the lower gut. These microbial communities hold immense significance in facilitating nutrient utilization by the host and contributing to the metabolic capabilities of the gut (as detailed in Table 2). This paradigm shift in research methodologies has provided us with a deeper understanding of the symbiotic connections between the microbial world and the host’s physiology in the context of the rumen and lower gut.

On the other hand, as the arena of bovine gut microbial ecology develops, multidisciplinary approaches must be used, coalescing host genomics and other omics-based techniques to understand the complex host-microbe network [83]. A universal approach that observes host/microbiome communications in both the rumen and the lower digestive tract is a prerequisite to harnessing the unlimited potential of the GI microbiome for sustainable ruminant production. Compared to that of the rumen, the lower-gut microbiota’s fundamental role(s) and its involvement in ruminant health and production are rarely reported [47]. Lower gut microbiota diverges in the composition according to intestinal segment, conferring the differences in the physical, chemical and biological conditions in each compartment. The dominant bacterial phyla in the rumen of cattle are typically *Firmicutes* and *Bacteroidetes*. Firmicutes are greatest in relative abundance in a predominantly forage-based diet, whereas *Bacteroidetes* are usually more abundant in diets consisting primarily of concentrate. *Firmicutes* and *Bacteroidetes* are then typically succeeded in a lot by *Proteobacteria*, *Tenericutes*, and *Actinobacteria*. *Prevotella* is most common at the genus level, potentially due to the wide range of functional capacities of species within *Prevotella* [96,97]. The jejunum is a site dominant in phyla *Firmicutes* (90%) and is a region of the post-ruminal protein, carbohydrate digestion and absorption [98]. The hindgut region consists of the cecum and colon majorly consists of microbial communities with similar functions, i.e., *Bacteroidetes*. *Firmicutes* [98]. A recent study by Vasco et al. in 2021 [99] demonstrated that the carriage of Shiga toxin-producing Escherichia coli (STEC) in cattle is influenced by highly diverse microbiota profiles, particularly those associated with diets dominated by forage. Furthermore, various factors impact the abundance of taxa linked to the shedding of STEC in dairy farms. These factors include dietary composition, the number of lactations, and warm temperatures. The formulation of starter feed is a critical consideration for effectively shaping the establishment of the gut microbiota and promoting rumen development. This significance stems from the fact that different dominant phyla within the microbiota necessitate specific fermentation substrates within the gut to derive the energy needed for their proliferation and colonization [100]. Notably, the prevalence of phyla in the cecal mucosa of adult goats fed starter feed and milk mirrors that of milk-fed lambs [33]. Dietary interventions have showcased their potential in influencing microbial dynamics and improving animal performance. For instance, supplementing the diets of calves with milk replacer, alfalfa hay, and starter feed has demonstrated enhancements in cecal volatile fatty acid (VFA) abundance and growth performance when compared to maternal grazing and nursing practices [101]. Moreover, recent research involving sheep indicated that inoculating newborn lambs with mature lyophilized rumen fluid led to improved starter feed digestibility and growth performance both during and post-weaning [102]. Similarly, the addition of a blend of essential oils and a prebiotic to ruminant diets has exhibited promising outcomes in enhancing growth, feed efficiency, nutrient digestibility, and immunity in calves over a 70-day period following birth [55,77]. These findings highlight the potential for dietary strategies to profoundly influence microbial composition, gut health, and overall animal performance in various ruminant species.

## 6. Lower-Gut Microbiome Influence the Immune System of the Host

Unlike the rumen, which lacks robust host immune mechanisms to maintain gut health, the lower gut regions exhibit a high degree of immune function. The mucosal immune system, comprising physical barriers (mucosal/epithelial layers) and chemical defenses (antimicrobial peptides, secretory IgA), as well as pattern-recognition receptors (such as toll-like receptors, TLRs) and a diverse array of immune cells, actively contributes to host defense [102,103]. The coevolution of numerous microorganisms inhabiting mammalian body surfaces and the immune system has resulted in a symbiotic relationship that is critical for host physiology. While many of these microbes perform essential functions for host health, they also pose a risk of causing harm if they deviate from their intended roles. The mammalian immune system plays a pivotal role in maintaining equilibrium with resident microbial communities, ensuring the mutualistic balance between hosts and microbes is upheld. Resident bacteria exert a significant influence on mammalian immunity, thereby recognizing the lower gut as a pivotal site for immune system development in monogastric animals [104].

There is mounting evidence indicating that the lower gut microbiome plays a role in establishing and maintaining immune system homeostasis in beef cattle [102,104], alongside its functions in feed digestion, energy production, and direct influence on gut health. Notably, the practice of starter feeding, a routine aspect of early-life calf management, exerts a significant influence on bacterial diversity and gene expression, particularly toll-like receptor genes (TLR10 and TLR2), associated with the efficiency of the host’s mucosal immune response in the lower gut [61]. In a related study, researchers found a close correlation between the total number of luminal and mucosa-associated bacteria in the small intestine of pre-weaned dairy calves and the expression of genes governing the host immune response [105]. However, this same group of researchers also discovered that the interplay between commensal gut microbes and the expression of specific host microRNAs (miRNAs) may contribute to immune system development in the gut of neonatal calves [106]. These findings collectively underscore the intricate interactions between the lower gut microbiome, host immune response and overall health status in cattle.

A recent study of functional metagenomic profiles obtained from the ileal tissue of *Lactobacillus*-dominant calves showed higher expression of genes involved in “leukocyte and lymphocyte chemotaxis” and the “cytokine/chemokine-mediated signaling pathway” [107]. Despite highly customized microbial populations, two different taxonomy-based clusters—*Lactobacillus* or *Bacteroides*—were identified. Amongst the clustered microbiomes, *Lactobacillus*-dominant ileal microbiomes had considerably lower abundances of *Bacteroides, Prevotella, Roseburia, Ruminococcus*, and *Veillonella* compared to the *Bacteroides*-dominated ileal microbiomes [107]. Indeed, recent research has shed light on the specific molecular mechanisms by which the lower gut microbiota influences immune system development in dairy calves. A study demonstrated that calves with a dominance of Lactobacillus in their ileum exhibited upregulated genes associated with cytokine/chemokine-mediated signaling pathways, as well as leukocyte and lymphocyte chemotaxis, pointing towards heightened inflammatory responses [107]. In contrast, Bacteroides-dominant calves showed upregulated genes linked to cell communication, cell adhesion, regulation of mitogen-activated protein kinase cascades, and response to various stimuli [107]. These findings suggest a dynamic interplay between the composition of the lower gut microbiota and the development of the calf’s immune system. It is proposed that nutritional manipulation strategies aimed at modulating the lower-gut microbiota could be leveraged to enhance the health and immunity of newborn calves. This insight underscores the potential for targeted interventions that harness the complex relationship between gut microbiota and host physiology to promote calf well-being.

An effective approach involves preventing the onset of hindgut acidosis, which arises when quickly digestible carbohydrates overflow into the hindgut for fermentation. This leads to the accumulation of acidic fermentation byproducts like short-chain fatty acids (SCFAs). This buildup is thought to lower the luminal pH, causing shifts in microbial composition and potential damage to the gut epithelium, thereby adversely impacting the health of the animals. While the connections between ruminal microorganisms and acidosis have been clearly established in previous research [108,109], the precise links between alterations in lower-gut microbiota and hindgut acidosis in ruminants remain not fully elucidated. As a result, modulating the communities in the lower gut holds promise for enhancing intestinal well-being in cattle [110].

Regarding adult cattle, the understanding of the lower gut and its functions remains limited in current research efforts. The maintenance of the host’s immune functionality and gut well-being requires energy consumption [111]. Consequently, challenges like stress and diseases can impede animal growth and the efficiency of production. The indigenous microorganisms present in the intestine, alongside external factors such as diet, play critical roles in maintaining immune equilibrium and reactivity, as well as influencing parameters like body weight and insulin resistance. Moreover, it is acknowledged that any disruption in metabolism over an extended period, such as obesity, can be linked to immune changes like inflammation. A more comprehensive investigation is warranted to thoroughly grasp the contributions of the lower-gut microbiome to both animal health and productive outcomes [110].

While much attention has been directed towards exploring the rumen microbiome in bovine gut studies, it is important not to underestimate the significance of the hindgut microbiome in terms of host well-being and production. This is particularly due to its vital role in feed digestion and the production of methane. Furthermore, the lower gut can also act as a reservoir for potential foodborne pathogens and the accumulation of nitrate wastes, underscoring the necessity of encompassing the entire gut microbiome in research endeavors aimed at enhancing animal health and productivity [112]. By acknowledging the contributions of the complete gut microbiome, researchers can take substantial steps toward advancing more efficient and sustainable practices within global livestock production.

## 7. Gut-Microbiota-Generated Metabolites in Animal Health

Metabolomics has brought about a transformative shift in the examination of intricate genetic, epigenetic, and environmental interplays by offering a tool to identify the ultimate outcomes of these interactions [113]. Metabolites have earned the moniker of “canaries” of the genome, given their sensitivity to shifts within the cellular milieu [112]. The mucosal surfaces of the gut, enveloped by the crucial gut epithelial layer, serve as a pivotal interface connecting animals with their surroundings. Recent studies have underscored the significance of maintaining gut cell homeostasis, influencing critical aspects of host physiology, encompassing development, metabolism, and immunity. Elie Metchnikoff’s century-old proposition regarding the gut’s influence on host well-being has found validation through contemporary research, showcasing how certain bacteria’s disruption of gut equilibrium can culminate in diseased states, attributed to the toxicity emanating from bacterial byproducts [114]. Within this context, metabolomics has significantly enriched our comprehension of the precise roles specific metabolites play in the intricate dynamics of gut health and disease.

The gut’s microbial community constitutes a dynamic system that takes shape and evolves post-birth, and in some instances, even prior to birth. This intricate system orchestrates health-related functions through the genetic coevolution linking hosts and gut microorganisms, yielding distinct nutritional symbioses like the production of short-chain fatty acids (SCFAs) and synthesis of vitamins, which may even be hereditary [115]. Hence, the integration of metabolomics into the study of the gut microbiome holds vital significance in deciphering the intricate interplay between microbial collectives and host well-being. In essence, the utilization of metabolomics has brought about a transformative shift in comprehending multifaceted genetic, epigenetic, and environmental interactions across various domains, including the gut microbiome. Recent investigations have substantiated Elie Metchnikoff’s conjecture concerning the gut’s impact on host vitality, highlighting the pivotal role of maintaining equilibrium in gut cell functions spanning various facets of host physiology. Through metabolomics, a more profound insight into the specific roles that distinct metabolites play in shaping gut health and disease has been attained, thus cementing its status as an invaluable instrument for delving into the complexities of the gut microbiome’s intricate workings.

The synthesis of antibodies constitutes a metabolically demanding procedure, the orchestration of which lies under the governance of the microbiota. This microbiota, a crucial component, assumes the role of a guardian by fending off pathogen colonization and preventing excessive growth of pathobionts, all the while nurturing a harmonious microbial community. The dynamics governing the microbiota’s adeptness in constraining pathogen expansion encompass multifaceted mechanisms such as competitive metabolic interplays, precise localization within distinct intestinal niches, and provocation of host immune reactions [113]. In parallel, pathogens have evolved strategies to elude commensal-mediated colonization resistance, further emphasizing the pivotal triad involving commensals, pathogens, and native pathobionts in orchestrating infection and disease. A more comprehensive comprehension of these intricate interactions might pave the way for innovative therapeutic avenues in the combat against infectious ailments.

The nutritional needs of individual commensal microorganisms constitute a pivotal determinant in shaping the composition and distribution of the microbiota. The establishment of microbial communities within mammalian hosts commences shortly after birth, and the bacterial makeup within both the oral cavity and the intestinal tract is initially rudimentary and comparable. However, the transition from maternal milk to fiber-rich sustenance after weaning triggers a significant alteration in bacterial composition. The small intestine, replete with mono- and disaccharides and amino acids, offers a conducive milieu for the proliferation and maturation of specific bacterial groups, including Proteobacteria and Lactobacillales. In the distal segment of the small intestine, where simple sugars are assimilated by host cells, the spectrum of energy sources available for bacterial cultivation undergoes a marked transformation, thereby ushering shifts in bacterial diversity. As one progresses beyond the ileocecal junction, the bulk of accessible carbohydrates is sourced from the diet or the host’s components (such as mucin and cellular remnants), constituting complex carbohydrates (polysaccharides) beyond the host’s digestive capacity. Consequently, Proteobacteria like Escherichia coli, hampered by their inability to metabolize polysaccharides, find themselves incapable of harnessing complex carbohydrates as an energy reservoir [112,116].

The configuration of the microbiota is profoundly shaped by the accessibility and allocation of nutrients throughout the gastrointestinal tract. Bacteroides and Clostridiales, equipped with enzymatic machinery proficient in dismantling host-resistant polysaccharides such as fibers and mucin, utilize these substrates as energy reservoirs, thereby claiming dominance within the expansive confines of the large intestine. In contrast, the prevalence of Proteobacteria and Lactobacillales is less pronounced in the colon [117]. Furthermore, even within the same phyla, the competence to digest polysaccharides can fluctuate markedly among bacterial cohorts, highlighting how the dietary composition of polysaccharides can exert discernible influence over the proportional abundance of bacterial species nestled within Bacteroidetes or Clostridiales. Nevertheless, the intricacies of the microbiota’s diversity across various animals cannot be solely attributed to dietary variations. Multiple investigations have unveiled shifts in microbiota configuration in animals afflicted by intestinal disorders, suggesting that perturbations in the supply of host-derived factors and nutrients underpin these changes in the context of diseased conditions [117].

Comprehending the breadth of bacterial diversity within the intestinal milieu necessitates a comprehensive exploration of the ecological interplays between commensal microorganisms and their intricate metabolic functionalities. Deeper investigations are imperative to elucidate the intricate mechanisms governing intra-commensal communication and to delineate the distinct metabolic roles undertaken by individual bacterial entities (Figure 5).

The crucial role of the microbiota and its metabolic byproducts in shaping animal physiology, encompassing health, nutrition, and production aspects, is widely acknowledged. A significant portion of metabolomics research in livestock, about 65%, is dedicated to exploring these realms [118]. Nevertheless, the realm of rumen metabolomics in beef cattle, especially untargeted approaches, remains relatively constrained in comparison to dairy cattle. A recent investigation by Sun et al. [119] delved into metabolites across four biofluids from dairy cows through NMR and GC techniques, underscoring the distinctive metabolomic profile of rumen fluid. In the context of beef cattle, efficient protein production stands as a paramount objective, necessitating the optimization of feed conversion efficiency. Gaining insights into the intricate interplay connecting the microbiome, feed conversion, and host physiology is pivotal for refining ruminant production strategies. For instance, variations in rumen fluid metabolomics were observed among steers displaying distinct feed efficiency levels. Notably, around 90 metabolites exhibited differences between high- and low-efficiency animals, largely implicated in fatty acid and amino acid metabolism [120]. Interestingly, divergences in plasma metabolites, notably pertaining to fatty acids, were also noted between animals characterized by differing feed efficiency. However, the concentrations of fatty acids demonstrated disparities between the rumen and plasma metabolomes, hinting at the involvement of additional factors influencing metabolite translocation from the rumen to the bloodstream [120].

The early lactation phase in dairy cows holds immense significance in terms of nutrition and management. During this critical period, a substantial portion of cereal grains is incorporated into the diet to fulfill the heightened energy and nutrient requisites for milk production and counterbalance any negative energy disparity. Cereal grains, abundant in starch, undergo rapid degradation in the rumen, giving rise to volatile fatty acids such as propionate, acetate, and butyrate [121]. Among these, propionate is harnessed by the liver to synthesize glucose, a pivotal energy source for oxidative processes. However, instances of reduced carbohydrate intake and propionate levels due to diminished appetite can prompt the liver to pivot towards employing non-esterified fatty acids (NEFAs) that traverse into the bloodstream for energy generation. Within the liver, NEFAs undergo oxidation into acetyl-CoA, which can either be completely oxidized through the tricarboxylic acid cycle (TCA), transformed into very-low-density lipoprotein (VLDL) and subsequently exported, or converted into triglycerides (TAG) and stored within the liver [122]. Nonetheless, the liver’s acetyl-CoA metabolism capacity is finite, and in scenarios where complete oxidation is unattainable, acetyl-CoA metamorphoses into ketone bodies, encompassing acetone, acetoacetate, and beta-hydroxybutyrate, which are subsequently excreted [123,124]. This phenomenon underscores the onset of the metabolic ailment known as ketosis (acetonemia) in dairy cows. In light of these dynamics, Saleem et al. [125] conducted a study demonstrating that cows fed with distinct diets, including rolled barley grain at varying proportions, exhibited marked alterations in the spectrum of both conventional and atypical rumen metabolites. This encompassed short-chain fatty acids, amino acids, ethanol, endotoxin, and monoamines.

Furthermore, a noteworthy augmentation in the levels of other biogenic amines like putrescine, cadaverine, and dimethylamine was identified in the ruminal fluid of dairy cows, concurrent with the escalation in MA concentrations, as explored in a study by Saleem et al. [125]. This finding is congruent with prior investigations that have illuminated the principal pathways of biogenic amine production in ruminants, including MA, cadaverine, and putrescine, which emanate from the decarboxylation of arginine, lysine, and arginine/ornithine [125,126]. The adoption of diets abundant in grains can elicit a decline in the ruminal pH, a pivotal factor steering the amino acid decarboxylase activity within the microbiota of the gastrointestinal tract’s lumen. Notably, the pH level holds the pivotal role of influencing the amino acid decarboxylase activity within the microbiota of the gastrointestinal tract’s lumen. In fact, amino acid decarboxylase activity has been underscored as a salient marker of acidotic conditions [116]. It is noteworthy that conditions like lameness and subacute ruminal acidosis (SARA) are commonly affiliated with the diminution in pH levels in cows [127].

Ethanolamine is a notable constituent of phospholipids within enterocyte membranes, originating from phosphatidylethanolamine [128]. The precise mechanism underpinning the release of ethanolamine into the rumen fluid of cows subjected to high-grain diets remains elusive. Nonetheless, there exists a conjecture that the alterations in microbial activity, coupled with modifications in the turnover of epithelial cells, and even the cell lysis of ruminal microbiota, could potentially contribute to the presence of ethanolamine in the rumen as a reaction to feedings rich in grains [128,129] (Table 3).

Emerging studies have unveiled a novel facet wherein ethanolamine can serve as a nitrogen source for pathogenic Gram-negative bacteria, exemplified by the enterohemorrhagic *E. coli* strain O157:H7. This adaptation provides these pathogens with a growth edge over other commensal microbiota [128]. Intriguingly, compounds like nicotinate or niacin, which are abundant in various feed sources, may encounter restrictions in accessibility, particularly in cereal-based diets for certain animal species [143]. Furthermore, the availability of niacin to both the host and rumen microbes could be constrained, especially when cattle are provided diets low in roughage content [143]. Many bacterial species capable of synthesizing nicotinate have been identified in rumen contents, including *Bacterium ruminicola* spp., *Bacteroides succinogenes*, *Ruminococcus flavefaciens*, *Lachnospira multiparus*, *Streptococcus bovis*, and *Butyrivibrio* [144]. Moreover, the presence of nicotinate in the rumen favors the deamination of amino acids, partly explaining the elevated ruminal levels of biogenic amines [144].

Traditionally, it has been widely accepted in reproductive science that the fetal environment remains sterile until after birth, with the acquisition of microbiota commencing subsequently. However, research conducted over three decades ago challenged this notion by confirming the presence of bacteria in amniotic fluid, even in the absence of ruptured membranes [130]. Subsequent investigations have further substantiated the notion of microbes being significantly present within the fetal-maternal unit, encompassing components like the placenta, amniotic fluid, and meconium [131]. This accumulation of evidence has given rise to the concept of a maternal-fetal microbial triad, incorporating a “holobiont” that exhibits a highly symbiotic relationship among its diverse constituents and experiences compromised functionality when its commensal microbiota encounters disturbances [145,146]. Inflammatory responses triggered by alterations in intrauterine microbial communities have been associated with adverse outcomes such as preterm birth and a spectrum of newborn conditions encompassing brain, lung, and eye ailments [131]. Consequently, the conventional belief in the newborn’s development taking place in a sterile environment is undergoing a transformation. The notion of in-utero colonization holds the potential to exert profound effects on the maturing mammalian host, influencing facets such as immune system development and metabolism, with potential implications for later-life diseases through epigenetic mechanisms [131]. The trajectory of microbial composition during and after birth is influenced by an array of factors, including delivery method, feeding type, and environmental exposures [132,133].

The metabolic activity of microorganisms holds significant importance in supporting host growth and lifespan. Recent advances in microbiome research have highlighted a noteworthy discovery: the microbiota does not merely evade the immune system to establish residence in the intestinal tract. Instead, a complex interaction has been unveiled, involving the indigenous microbiome, the intestinal epithelium, and local immune cells. Within this intricate system, all participants actively contribute to maintaining gastrointestinal equilibrium. Central to this dynamic are metabolites derived from bacteria, which serve as essential signals continuously enhancing the integrity of the epithelial barrier and bolstering immune functions [134].

## 8. Future Prospective

Understanding the impact of the microbiome on production traits: Research can focus on understanding how the microbiome of the rumen and other digestive organs influences production traits such as feed efficiency, weight gain, and milk yield. By better understanding these relationships, it may be possible to develop management strategies to optimize microbiome function and improve production efficiency.Developing microbial interventions for improved animal health: Similar to human medicine, microbial interventions could be developed to promote health and prevent disease in cattle. For example, probiotics or microbial supplements could be used to promote beneficial microorganisms or microbial therapies could be used to treat infections or other health conditions.Exploring the role of the microbiome in animal behavior: Emerging research suggests that the microbiome may play a role in regulating animal behavior, potentially influencing stress response, social behavior, and other traits. Investigating these relationships in cattle could provide new insights into how the microbiome influences animal welfare and production.Investigating the impact of management practices on the microbiome: Management practices such as feed composition, housing conditions, and antibiotic use may all influence the composition and function of the cattle microbiome. Understanding these relationships can help identify best practices for managing the microbiome and improving animal health and productivity.Developing precision management tools for the microbiome: Advances in technology are enabling increasingly precise analysis of the microbiome which could, in turn, enable more targeted and personalized management strategies. For example, microbial profiling could be used to identify individual animals or herds with particular microbiome profiles, which could be managed in a more targeted manner to optimize production and health outcomes.

## 9. Limitation of the Study Topic

One major limitation of research in the field of cattle microbiome is the limited understanding of the microbial dynamics and interactions within the rumen. The rumen is a highly complex and dynamic environment, and microbial communities within it can change rapidly in response to various factors such as diet, age, and management practices. Despite advances in sequencing technologies, there are still challenges in accurately identifying and characterizing microbial taxa within the rumen. This is due to the high diversity of microbial species present, as well as limitations in the reference databases used for taxonomic assignment.

Another limitation is the lack of standardization in sampling and analysis methods. Different sampling and processing methods can lead to variations in the results obtained, making it difficult to compare data across studies. Additionally, there is a need for larger-scale longitudinal studies to understand how microbial communities change over time in response to different factors, as well as how these changes impact animal health and productivity.

Lastly, the practical application of microbiome research in the livestock industry is still in its early stages. While there is potential for microbiome-based interventions to improve animal health and productivity, there is a need for more research to identify effective strategies for manipulating the microbiome. Additionally, there are regulatory and economic considerations that need to be taken into account when developing microbiome-based interventions for use in the livestock industry.

## 10. Conclusions

In summary, the gut microbiome holds a pivotal role in maintaining the health and efficiency of animals, with various elements contributing to ruminant performance characteristics. These include the rumen microbiome, host physiology, and external factors like diet and management practices. Studies have revealed connections between different metabolites and key production-related traits in cattle, particularly those tied to intermediary metabolism. While the potential of the gut microbiome to enhance livestock production on a global scale is substantial, there remains much to learn about its manipulability, the factors shaping its establishment, and the consequent physiological impacts on the host. Furthermore, gastrointestinal microbes play a significant role in fostering immune system development, generating bioactive compounds, extracting energy, and a myriad of other processes that collectively contribute to overall health and well-being in both animals and humans. This interaction helps mitigate various stressors, whether they are physiological or psychological in nature. While the prospect of manipulating the gut microbiome to enhance animal health and productivity is promising, the research in this field also presents limitations, including the current lack of knowledge about potential long-term effects and unintended outcomes from such manipulations.

In conclusion, the gut microbiome acts as a crucial facilitator of nutrient utilization in animals, including cattle, and holds potential for further investigation to fully comprehend its intricate interplay with host physiology and external influences. This understanding is crucial for developing more effective strategies aimed at improving animal health and productivity, thus contributing to meeting the global demand for livestock production.

## Figures and Tables

**Figure 1 biology-12-01200-f001:**
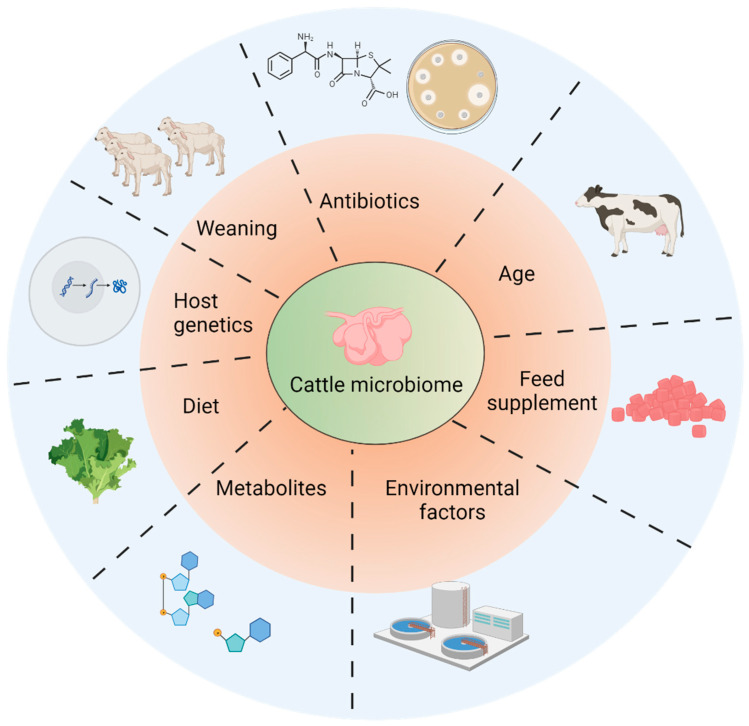
Factors affecting the establishment and development of microbiota throughout the gastrointestinal tract (GIT) in cattle are multifaceted and can significantly impact the health and productivity of the animal. The composition and abundance of microbiota in the rumen and lower gut of cattle are influenced by several factors, including diet, host genetics, age, environmental conditions, metabolites, management practices, and exposure to antimicrobials.

**Figure 2 biology-12-01200-f002:**
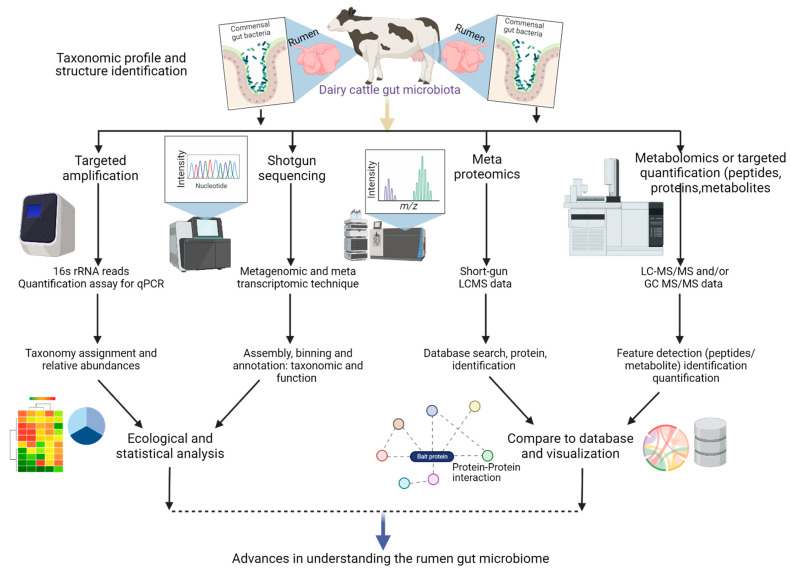
The figure represents a schematic presentation of the microbiota in farm animals, particularly focusing on the metagenomics, proteomics, metabolomics analysis of the rumen microbiome. It highlights the high-throughput analysis workflow and the recent advances in understanding the rumen microbiome. The microbiome in cattle is a complex community of microorganisms that plays a crucial role in feed digestion, fermentation, and nutrient absorption, influencing the host’s health and productivity. The use of molecular biology techniques, particularly metagenomics proteomics, and metabolomics has expanded the knowledge and understanding of the rumen microbiome’s complexity and diversity. These techniques enable the identification and characterization of the microbial community structure, functional genes, and metabolic pathways in the rumen. The figure also emphasizes the importance of a multidisciplinary approach that integrates omics-based techniques to understand the complex host-microbe network in the rumen.

**Figure 3 biology-12-01200-f003:**
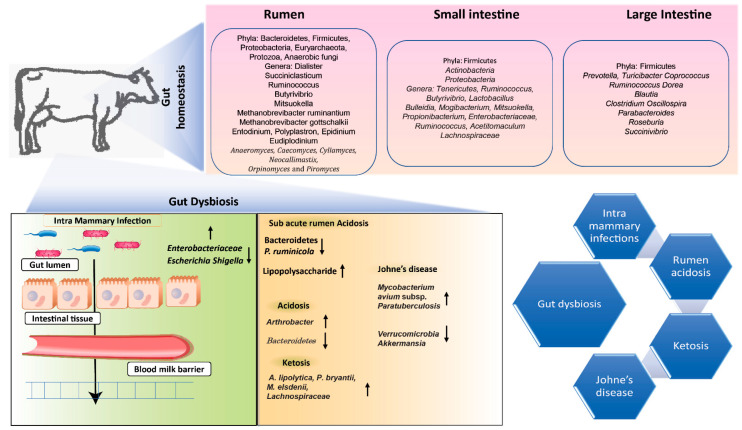
Role of Microbiome in Maintaining Homeostasis and Its Implication in Diseases Such as Intra Mammary Infections, Dysbiosis, Rumen Acidosis, Johne’s Disease, and Ketosis.

**Figure 4 biology-12-01200-f004:**
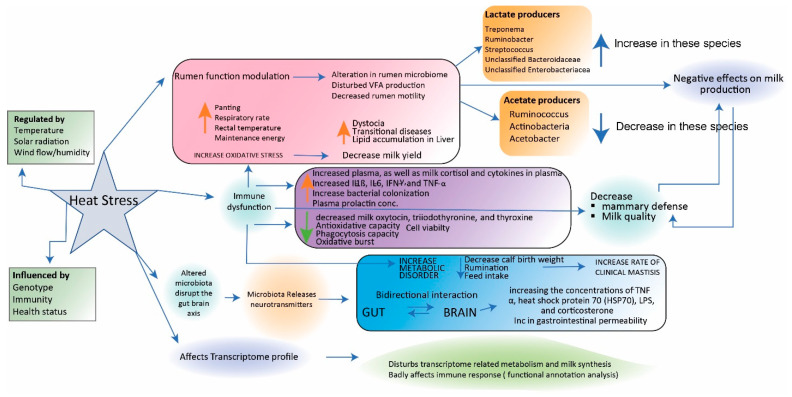
Effects of heat stress on dairy cattle production and immune system: An exploration of the impacts of high temperatures on milk yield, reproductive performance, and health outcomes in cattle. The role of the immune system in mitigating heat stress and the potential influence of microbiome on immune response.

**Figure 5 biology-12-01200-f005:**
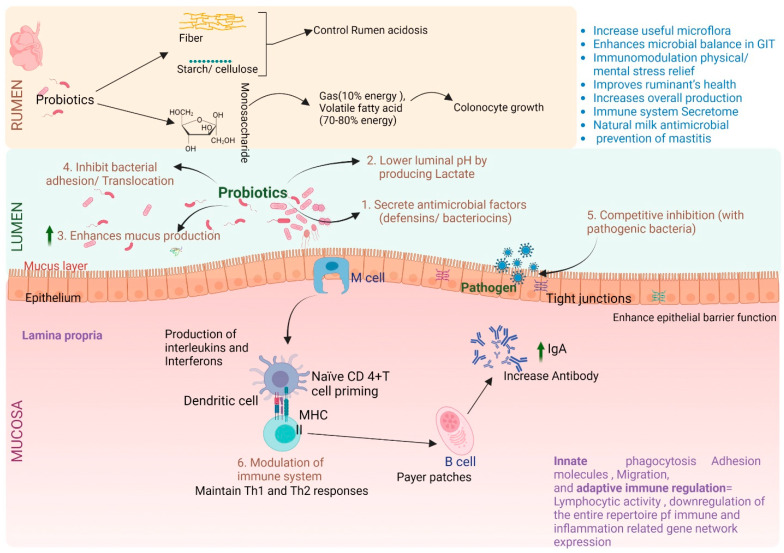
The role of innate and adaptive immune regulation in the modulation of phagocytosis, adhesion molecules, and migration in relation to microbiota. This includes lymphocytic activity and downregulation of the entire repertoire of immune and inflammation-related gene network expression.

**Table 1 biology-12-01200-t001:** Exploring the multifaceted influences on rumen microbiota composition: comparative analysis of various models and analytical techniques revealing key factors and insights.

Factors	Model	Technology	Results	References
Age	Bovine	454 tag-encoded amplicon pyrosequencing	Between high residual feed intake groups and low residual feed intake groups, the diversity and within-group similarity increase with age, and each group has its own distinct microbiota	[30]
	Calf	16S rRNA gene sequencing, whole genome shotgun approach	Rumen microbiota of preruminant calves displays compositional heterogeneity, but functional classes between the 2 age groups (14-day-old calves and 42-day-old calves) are similar	[31]
	Calf	16S rRNA gene amplicon sequencing and functional composition prediction.	The fecal microbial community has a great alteration within the time after weaning in both cattle and yak calves, but cattle showed a larger change.	[32]
	Bovine	16S rRNA gene amplicon sequencing and microbial diversity analysis	Prevotella was strongly correlated with methanobrevibacter in heifers; however, in older cows (96–120 months) this association was replaced by a correlation between *Succinivibrio* and *Methanobrevibacter*	[33]
Diet	Bovine	Metabolomics	Roughage type can significantly influence the ruminal microbial metabolome, especially organic acids, amines, and amino acids	[34]
Feed efficiency	Bovine	16S rRNA gene sequencing, shotgun DNA sequencing	*Megasphaera elsdenii* is enriched in the rumen of the feed-efficient group; *Methanobrevibacter* was diminished	[35]
	Bovine	Metatranscriptomics	*Lachnospiraceae*, *Lactobacillaceae*, and *Veillonellaceae* are more abundant in low-feed efficiency animals; *Methanomassiliicoccales* was diminished	[36]
	Bovine	metagenomics, metatranscriptomics, and metabolomics	Rumen of HiEf animals, *Selenomonas* and some species of *Succinivibrionaceae* might interact positively with each other and play an important role as keystone bacteria.	[37]
	Bovine	16SrRNA Gene Sequencing	Host-associated microbial interactions differed within each breed depending on the feeding system, which indicated that breed-specific feeding systems should be taken into account for farm management	[38]
Genetics	Bovine	SNP-based heritability estimates and 16S rRNA gene sequencing	Host genetic variation is associated with specific microbes	[39]
	Bovine	Metagenomics	Host genetics shapes the microbiome Inoculation with bison rumen contents alters the cattle rumen microbiome and metabolism	[40,41]
	Bovine	High throughput sequencing	Rumen microbial features are heritable and could be influenced by host genetics, highlighting a potential to manipulate and obtain a desirable and efficient rumen microbiota using genetic selection and breeding.	[42]
	Bovine	16S rRNA gene sequencing	Single-nucleotide polymorphisms found in cattle genomes and corresponding rumen bacterial community composition, annotated genes associated suggest the associations observed are directed toward selective absorption of volatile fatty acids from the rumen to increase energy availability to the host	[43]

**Table 2 biology-12-01200-t002:** Comprehensive overview of high-throughput studies identifying distinct types of bacterial species: providing in-depth insights into their specific functions, potential, and the isolation of resultant end products.

Bacterial Species	End Product	Function	Reference
*Fibrobacter succinogens*, *Clostridium longisporum*, *Clostridium cellobioparum*	Acetate, Formate, Ethanol, propionate	Degrade the cellulose into smaller oligo/disaccharides	[84]
*Butyrivibrio fibrisolvens*	Saturated fatty acids and conjugated linoleic acids	Lipolysis and biodehydrogenation	[85]
*Prevotella ruminicola*, *Ruminobacter amylophilus*, *Streptococcus Bovis*		Hydrolysis of starch	[86]
*Anaerovibrio lipolytica*	Acetate and propionate	Lipolytic activity of rumen contents of cattle	[87]
*Lachnospira multiparus*, *Treponema saccharophilum*	Acetate and formate	Fermentation of pectin and glucose	[88]
*Methanobrevibacter* spp.	Methane	Methane emissions	[89]
*Succinivibrio* sp. *Lactobacillus* sp.	Lactate, Acetate, Fumarate, Succinate	Lactate, Acetate, Fumarate, Succinate	[84,90]
*Prevotella ruminicola*, *Eubacterium uniformis*, *Eubacterium xylanophilum*	Acetate, Formate, Ethanol, propionate	Xylan consumption	[88,91]
*Ruminococcaceae*, *Clostridium*, *Turicibacter* and unclassified *Peptostreptococcaceae*	Volatile fatty acid	Degradation of starch and fiber	[45]
*Ruminobacter amylophilus*, *Prevotella* sp., *Clostridium bifermentans*	Amino acids, nitrogen	Starch hydrolysis	[91]
*Entodiniomorphs*, *Eudiplodinium*, *Dasytricha*, *Diplodinium*, *Metadinium*, *Ophryoscolex*, and *Ostracodinium*	Methane	Support methanogenesis, Engulf and digest a wide range of bacteria	[92,93]
*Isotricha* *Dasytricha* *Charonina*	Glucose, Fructose, Galcturonic acid	Fermentation of Starch, pectin, soluble sugars, proteins	[94]
*Megasphaera elsdenii*	Ammonia and CO_2_	Acts upon lactate (end product of bacterial fermentation)	[84,90]
*Entodinium*, *Diplodinium*, *Eudiplodinium*, *Ostracodinium*, *Metadinium*, *Polyplastron*, *Ophryoscolex*, *Epidinium*	Glucose, Xylose	Hydrolyze structural polysaccharides	[95]
*Eubacterium oxidoreducens*, *Streptococcus Caprinus*	Lactate, Acetate, Fumarate, Succinate	Degradation of tannin	[83]

**Table 3 biology-12-01200-t003:** Overview of Metabolites Derived from Microbiota, their Associated Model Organisms, Roles in Diseases/Infection Control, and Mechanisms of Action.

Metabolites	Model Organism	Diseases/Infection Control	Mode of Action	Reference
SCFA	Cow	Mastitis	Improve condition by conserving mucosal barrier integrity, restoring blood-milk barrier and prevent pathogens and their metabolites intrusion, maintain immune homeostasis	[130]
Ursodeoxycholic acid	Neonatal dairy calves	ESBL-EAEC-induced clinical symptoms and colitis	Show antibacterial action, inhibit proinflammatory effect, and decreased cell integrity loss	[131]
Deoxycholic acid	Chicken	*Campylobacter jejuni* colonization	Prevent infection by modulating microbiota composition	[132]
Butyrate	Cattle	Subacute ruminal acidosis (SARA)	Prevent SARA by increasing ruminal pH	[133]
SCFA	Pig	Intestinal Barrier Function	Improve gut health by regulating IL-22 production	[134]
Indole derivatives (3-indoleacrylic acid, kynurenic acid)	Simmental cattle	Daily weight gain	Enhance body weight by regulating intestinal homeostasis, gut barrier function and immune system	[135]
Putrescine	Pig	Diarrhea	Alleviate diarrhea by enhancing anti-inflammatory function and suppressing inflammatory response	[136]
volatile fatty acids (VFAs)	Dairy Cow	Subacute ruminal acidosis (SARA)	Cause acidosis by decreasing rumen pH below physiological range	[137]
Butyrate	Pig	lipopolysaccharide (LPS)-induced colitis	Protect against colitis by diminishing secretion of pro-inflammatory cytokines (IL)-1β, IL-6, tumor necrosis factor (TNF)α, IL-8, and IL-12	[136]
Leukotriene B4	Cow	Clinical mastitis	Cause mastitis by increasing inflammatory response	[138]
Histamine	Dairy Cow	Subacute ruminal acidosis (SARA)	Cause SARA diseases by activating NF-kb and mTOR signaling followed by mammary inflammtion	[139]
LPS, lactate	Dairy Cow	Laminitis	Disturb rumen environment by decreasing rumen pH	[140]
5-Hydroxymethyl-2-furancarboxaldehyde	Lactating Holstein dairy cows	Clinical Mastitis	Cause mastitis by enhancing production of pro-inflammatory factors, such as TNF-α and IL-1β	[138]
δ-aminolevulinic acid	Dairy Calves	Diarrhea	Cause diarrhea by gut dysbiosis	[141]
Non-esterified fatty acid (NEFA), and β-hydroxybutyric acid (BHBA)	Dairy Cow	Left Displaced Abomasum	Cause LDA by increased Ketosis and negative energy balance (NEB) prepartum	[142]

## Data Availability

Not applicable.

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
