# Peer review of "Integrating Omics Technologies for a Comprehensive Understanding of the Microbiome and Its Impact on Cattle Production"

_biology, 2023, doi:10.3390/biology12091200_

Round 1

Reviewer 1 Report

The abstract should be better elaborated. Authors should better synthesize the ideas addressed in the text. It is clear that some sentences are taken in full from the text. Keywords must be presented in alphabetical order and must not repeat the words present in the title. References must be checked and adjusted according to the journal's guidelines. Some references lack information

Author Response

We sincerely appreciate the reviewer's feedback and assessment of the abstract. We recognize that the abstract can benefit from more elaboration and synthesis of the ideas presented in the manuscript. We have taken this feedback to heart and have revisited the abstract, working on presenting a more concise yet comprehensive summary of the key points discussed in the review article.

Regarding the keywords, we apologize for any oversight in their organization and repetition. We have reviewed and rearranged the keywords to ensure they are presented in alphabetical order and are distinct from the words used in the title.

Furthermore, we appreciate your pointing out the issues with the references. We have meticulously reviewed each reference to ensure they conform to the journal's guidelines and have provided complete information for all references. Any lacking information has been rectified to ensure accuracy and compliance.

Once again, we sincerely thank the reviewer for their valuable insights, and we have taken proactive steps to address these points and enhance the quality and presentation of our manuscript accordingly.

Reviewer 2 Report

This review covers a very interesting topic that has been the subject of a lot of research, namely the intestinal microbiome of cattle. Most studies on this topic are descriptive studies and moving from describing microbial composition to understanding host-microbiota interactions and then to concrete applications is a big step. Much research looks at microbial composition but without further functional investigations of these microbiota. Integrated studies combining different technologies are necessary to make the step towards a better understanding of the rumen function. The title of this review aspires to make this integration. Yet it is rather a listing of different studies that make use of different technologies rather than a step towards integration and interaction between omics technologies.

Major comments

The review is very comprehensive but is not very structured so it contains a lot of repetitions. The authors have to rethink the structural outline of the review in order to avoid repetition and to increase readability. 

The quality of the figures and tables is insufficient and these clearly need to be reworked. The tables are a compilation of studies often brought together for an unclear reason, with some studies appearing irrelevant to the intended objective. The figures are cluttered, with different fonts and cluttered layout.

The title talks about a comprehensive understanding of the rumen microbiome, but the article uses gastrointestinal system, rumen, gut and lower gut interchangeably. This is partly due to a lack of focus on the rumen, but also to confusing nomenclature.

Minor comments:

L19: Optimizing is not the best word. Optimizing can have any meaning depending on what you think is optimal.

L33-35: repetition of what was said in all previous lines

L41: what is the definition of an entity?

L42: waste management is maybe not the best example in this context?

L49: check reference notation style

L72-75: repetition of line 50-52. More repetitions occur in the rest of the paper. It is the authors responsibility to avoid these repetitions and make the paper more focused and coherent.

L130: Advances in genome-wide sequencing ap-130 proaches, such as metagenomics, metatranscriptomics, and metaproteomics AND METABOLIMICS (cfr line 134).

L136: RNA-seq is another name for metatranscriptomics and not for metagenomics. Likely refer the authors are confused by 16S rRNA sequencing which is in fact DNA sequencing of the 16S rRNA gene.

L207: Rumen Archaea

L245: body composition?

L296: The composition of microbial communities in the rumen is dependent on several factors, including breed, age, external environment, diet, and nutrition (Table 1): repetition of what is already described in other parts of the manuscript.

L313: Dominant bacteria Prevotella might have been stimulated by fiber-rich dietary supplements (Figure 4): I do not see this information in Figure 4

L337: “Additionally, the housing system's imbalance in terms of roughage to concentrated feed, frequent and quick changes in food doses… “. What is meant by housing system’s imbalance? The examples are more related to feed management than to housing?

L 341: livelihoods = environments?

L511: this section is about metabolites although metabolomics is at first sight and based on Figure 1 not part of the scope of this paper?

Figure 1: host genetics Is not indicated in the overview but is mentioned in the legend.

Figure 2: Targeted quantification of proteins by LC-MC/MS or GC?

Figure 2: There is a discrepancy between the figure and the figure legend: In this figure legend but also on other locations in the manuscript gut, rumen and digestive tract are used interchangeably without clarifying whether they have a more specific meaning? In the figure, only the rumen is shown, but in the figure legend, lower digestive tract is also described. In this figure legend, integration of the host genome is mentioned but it is not clear to me how this is integrated in the workflow.

Figure 3: Lay-out

Table 1: The results are briefly described but sometimes not very informative, for example

-        Ref 36: group is not specified

-        Ref 29: age is not specified

-        Ref 60: DGGE is not mentioned in the rest of the paper so this reference is out of the scope of the review

-        Breed is not in the table as a factor

Table: 1 check the use of capitals and punctuations

Table 2: The type of flora is a mix of functional classes such as Cellulolytic bacteria but also the fylum Firmicultes comprising several functional classes. Proteolytic bacteria have protease activity and not amylase activity?

Author Response

This review covers a very interesting topic that has been the subject of a lot of research, namely the intestinal microbiome of cattle. Most studies on this topic are descriptive studies and moving from describing microbial composition to understanding host-microbiota interactions and then to concrete applications is a big step. Much research looks at microbial composition but without further functional investigations of these microbiota. Integrated studies combining different technologies are necessary to make the step towards a better understanding of the rumen function. The title of this review aspires to make this integration. Yet it is rather a listing of different studies that make use of different technologies rather than a step towards integration and interaction between omics technologies.

Ans: Thank you for your insightful comment and valuable feedback on our review paper. We appreciate your recognition of the importance of the intestinal microbiome in cattle and the progression of research from descriptive studies to understanding host-microbiota interactions and practical applications.

Major comments

The review is very comprehensive but is not very structured so it contains a lot of repetitions. The authors have to rethink the structural outline of the review in order to avoid repetition and to increase readability. 

Ans: Thank you for taking the time to provide valuable feedback on our review paper. We sincerely appreciate your thoughtful evaluation of our work. Upon careful consideration of your comment, we recognize the need to improve overall readability. Your insight has guided us in reevaluating the outline and content flow to ensure a more coherent and engaging presentation. We highly value your input and have taken the necessary steps to address the concerns you raised. The revised version of the review now offers a clearer structure, reducing repetition and enhancing the flow of information for the benefit of our readers.

The quality of the figures and tables is insufficient and these clearly need to be reworked. The tables are a compilation of studies often brought together for an unclear reason, with some studies appearing irrelevant to the intended objective. The figures are cluttered, with different fonts and cluttered layout.

Ans: Thank you for your valuable feedback regarding the quality of the figures and tables in our manuscript. We sincerely appreciate your insights and have taken your comments into serious consideration. To address your concerns, we have made significant improvements to the figures. We have ensured that the figures are now presented in a clear and organized manner, with consistent fonts and layout to enhance readability and visual appeal. Regarding the tables, we apologize for any confusion caused by the previous version. In the revised manuscript, we have thoroughly reviewed and updated the table captions to provide a clearer explanation of the purpose and relevance of each compilation of studies. This will help ensure that the tables contribute effectively to the intended objectives of the review. Moreover, to enhance the visual quality of the figures, we have provided high-resolution images at 300 dpi to ensure the clarity and accuracy of the visual representations.

The title talks about a comprehensive understanding of the rumen microbiome, but the article uses gastrointestinal system, rumen, gut and lower gut interchangeably. This is partly due to a lack of focus on the rumen, but also to confusing nomenclature.

Ans: We have considered your suggestion and removed the term "Rumen" from the title to better align with the content of the article and provide a more focused and accurate representation of the scope. We believe that this adjustment will enhance the readability and relevance of the title, making it more reflective of the comprehensive understanding of the microbiome that our article aims to provide.

Minor comments:

L19: Optimizing is not the best word. Optimizing can have any meaning depending on what you think is optimal.

Ans: Indeed optimization here can have different meanings therefore, we have replaced this word with enhancing that significantly explaining the crucial need of promoting the rumen production systems.

L33-35: repetition of what was said in all previous lines

Ans: In response to the reviewer's feedback, we have made revisions to address the perceived repetition in describing the influence and advantages of studying the cattle rumen microbiome in relation to animal health and production. The suggested changes have been incorporated to ensure a more concise and coherent presentation of these points. We appreciate the reviewer's insights and believe that the modifications will enhance the clarity and flow of the manuscript. “Furthermore, detailed analysis of microbiome and its associated effects on the cattle production system greatly contributes to more sustainable and resilient food production system and in turn mitigate the negative impact on environment.”

L41: what is the definition of an entity?

Ans: In the context of microbial entities, the term "entity" denotes a discrete and distinguishable unit of microorganisms with distinct characteristics. In essence, a microbial entity encompasses one or more microorganisms and can be defined by factors such as taxonomic classification, functional attributes, and ecological roles. This concept highlights the individuality and separateness of microbial units within a broader microbial community.

L42: waste management is maybe not the best example in this context?

Ans: In this context, the authors delineated the diverse functions of the microbial community, highlighting its substantial involvement in the decomposition and transformation of organic matter. These microorganisms exhibit the capability to break down intricate organic substances into simpler forms, facilitating the recycling of organic materials. Notable instances of these processes encompass decomposition, organic matter degradation, composting, and anaerobic digestion.

L49: check reference notation style

Ans: The reference notation style has been amended and rectified. References are now positioned before the concluding full stop, rather than after it.

L72-75: repetition of line 50-52. More repetitions occur in the rest of the paper. It is the authors responsibility to avoid these repetitions and make the paper more focused and coherent.

Ans: Thank you for your comment. Your input is greatly appreciated, and we have taken it into consideration. We have made the necessary revisions to the manuscript to address the repetitions and improve the overall focus and coherence of the paper. Your feedback has been valuable in enhancing the quality of our work.

L130: Advances in genome-wide sequencing ap-130 proaches, such as metagenomics, metatranscriptomics, and metaproteomics AND METABOLIMICS (cfr line 134).

Ans: We have also incorporated the term "metabolomics" into the text.

L136: RNA-seq is another name for metatranscriptomics and not for metagenomics. Likely refer the authors are confused by 16S rRNA sequencing which is in fact DNA sequencing of the 16S rRNA gene.

Ans: Thank you for pointing out the clarification regarding the terms used. We appreciate your input and have made the necessary correction to accurately reflect the distinction between RNA-seq and metagenomics in the context of our manuscript. “The RNA sequencing refers to the transcriptomics which is different from the metatranscriptomics. The transcriptomics refers to the sequencing of RNA from a specific microbial species and it represents comprehensive set of RNAs encoded by and a particular organisms genome. On the other hand, metatranscriptomics involves the study of gene expression patterns across multiple organisms in a microbial community. Metatranscriptomics analyzes the total RNA extracted from complex microbial communities, including mixed cultures or environmental samples. It provides a holistic view of the gene expression landscape within a microbial community, offering insights into the functional potential and activities of diverse microorganisms present. Metagenomics on the other hand, is used to determine the microbial diversity and thus taxonomic profiling of microbial community. (Aguiar-Pulido, Vanessa, et al. 2016). “

Citation:  Aguiar-Pulido V, Huang W, Suarez-Ulloa V, Cickovski T, Mathee K, Narasimhan G. Metagenomics, Metatranscriptomics, and Metabolomics Approaches for Microbiome Analysis: Supplementary Issue: Bioinformatics Methods and Applications for Big Metagenomics Data. Evolutionary Bioinformatics. 2016;12s1. doi:10.4137/EBO.S36436

L207: Rumen Archaea

Ans: The spelling of Archae is replaced by Archaea

L245: body composition?

Ans: The term "bodily composition" in this context pertains to the modifications induced by changes in the host body's ecosystem, encompassing alterations in tissues, organs, and overall structural attributes. In our manuscript, we aim to underscore the significance of the modified rumen microbiome in influencing the composition of the host's body.

L296: The composition of microbial communities in the rumen is dependent on several factors, including breed, age, external environment, diet, and nutrition (Table 1): repetition of what is already described in other parts of the manuscript.

Ans: Thank you for your valuable feedback. We have carefully considered your observation regarding the repetition in our manuscript. We acknowledge that the factors influencing the composition of microbial communities in the rumen, including breed, age, external environment, diet, and nutrition, have been discussed in multiple sections of the paper. In response to your feedback, we have made the necessary revisions to eliminate redundancy and ensure a more streamlined and focused presentation of these aspects. We appreciate your diligence in reviewing our work and for bringing this matter to our attention.

L313: Dominant bacteria Prevotella might have been stimulated by fiber-rich dietary supplements (Figure 4): I do not see this information in Figure 4

Ans: We apologize for the oversight regarding the citation of Figure 4. In the revised manuscript, we have corrected the in-text citation for Figure 4 accordingly. Thank you for bringing this to our attention.

L337: “Additionally, the housing system's imbalance in terms of roughage to concentrated feed, frequent and quick changes in food doses… “. What is meant by housing system’s imbalance? The examples are more related to feed management than to housing?

Ans: Thank you for providing further clarity on the concept of "housing of the animals." We appreciate your input and acknowledge the multi-faceted nature of this term. It encompasses not only the types of feed provided and the rearing facilities, but also extends to the overall living environment, including factors such as herd size and volume. This comprehensive definition adds depth to our understanding and will be incorporated into the manuscript to enhance the accuracy of our content. Your contribution is valuable in refining the discussion on this topic.

L 341: livelihoods = environments?

Ans: Certainly, the term "livelihood" within our manuscript pertains to the diverse environmental conditions in which cattle are raised, consequently influencing their rumen microbiome. Geographical variations give rise to alterations in feed composition, encompassing both chemical and botanical distinctions. Notably, the environment in which animals reside has been empirically shown to exert an impact on their rumen microbiome composition. This environmental dimension is an integral aspect of our analysis, contributing to a more comprehensive understanding of the factors influencing rumen microbial communities. Your clarification further enriches our discussion by emphasizing the intricate interplay between the cattle's habitat and their rumen microbiome composition.

L511: this section is about metabolites although metabolomics is at first sight and based on Figure 1 not part of the scope of this paper?

Ans: Thank you for bringing this to our attention. We have taken your feedback into consideration, and indeed, the discussion on metabolomics and metabolites is a part of our manuscript. We have made the necessary adjustments by updating the figure 1 in the revised manuscript to include a specific section on metabolites. Consequently, in Section 7 of the revised manuscript, we comprehensively address the generation and function of metabolites within the cattle-associated microbiome. Your observation has enabled us to enhance the clarity and completeness of our discussion on this crucial aspect of our research.

Figure 1: host genetics Is not indicated in the overview but is mentioned in the legend.

Ans: Thank you for pointing out. We have updated the figure 1 in the revised manuscript.

Figure 2: Targeted quantification of proteins by LC-MC/MS or GC?

Ans: Thank you for bringing this to our attention. We appreciate your feedback, and we have made the necessary revisions to the figure in the revised manuscript. The updated figure now illustrates the targeted quantification of peptides, proteins, and metabolites through the utilization of LC-MS/MS or GC techniques. These refinements enhance the clarity and accuracy of our depiction of these analytical methodologies.

Figure 2: There is a discrepancy between the figure and the figure legend: In this figure legend but also on other locations in the manuscript gut, rumen and digestive tract are used interchangeably without clarifying whether they have a more specific meaning? In the figure, only the rumen is shown, but in the figure legend, lower digestive tract is also described. In this figure legend, integration of the host genome is mentioned but it is not clear to me how this is integrated in the workflow.

Ans: Thank you for bringing this to our attention. We greatly value your feedback, and we have taken the necessary steps to ensure clarity in the terminology used for "gut," "rumen," and "digestive tract" in the legend of Figure 2, as well as throughout the entire revised manuscript. With these updates, any confusion related to the host genome has been effectively addressed and eliminated from the updated figure.

Figure 3: Lay-out

Ans: Thank you for pointing out. We have updated the layout of the figure 3 in the revised manuscript.

Table 1: The results are briefly described but sometimes not very informative, for example

-        Ref 36: group is not specified

Ans: Thank you for pointing out, it is about high residual feed intake groups and low residual feed intake groups. We have updated the same information in the revised manuscript.

-        Ref 39: age is not specified

Ans: Thank you for pointing out, the age group is 14-day-old calves and 42-day-old calves.

-        Ref 60: DGGE is not mentioned in the rest of the paper so this reference is out of the scope of the review

Ans: Thank you for pointing out, we have removed the reference from the table.

-        Breed is not in the table as a factor

Ans: Thank you for pointing out, we kept Bovine and calf as the model for simplification of the tabular information

Table: 1 check the use of capitals and punctuations

Ans: Thank you, we have made revisions to ensure consistency in the text.

Table 2: The type of flora is a mix of functional classes such as Cellulolytic bacteria but also the fylum Firmicultes comprising several functional classes. Proteolytic bacteria have protease activity and not amylase activity?

Ans: We appreciate your observation. We have taken the necessary step of removing the "Type of flora" column from Table 2 to enhance clarity and improve the presentation of information.

Reviewer 3 Report

In my opinion the manuscript is well written and apport new and interesting information. For these reasons I suggest to accept the manuiscript

Author Response

We greatly appreciate your positive feedback and the time you've taken to review our manuscript. Your kind words and support are encouraging to us. We are pleased that you found the manuscript to be well-written and informative, and we are glad to hear that you recommend its acceptance. Your input is invaluable to us, and we are grateful for your endorsement of our work. Thank you for your thoughtful review, and we look forward to any further comments or suggestions you may have.